# Spectral properties of Lévy Rosenzweig-Porter model via supersymmetric approach

Elizaveta Safonova[1,2], Mikhail Feigelman[1] and Vladimir Kravtsov[3]

**1** Nanocenter CENN, Ljubljana, Slovenia
**2** Department of Physics, University of Ljubljana, Slovenia
**3** ICTP, Trieste, Italy

September 4, 2024

## Abstract

By using the Efetov's super-symmetric formalism we computed analytically the mean spectral density $\rho(E)$ for the Lévy and the Lévy -Rosenzweig-Porter random matrices which off-diagonal elements are strongly non-Gaussian with power-law tails. This makes the standard Hubbard-Stratonovich transformation inapplicable to such problems. We used, instead, the functional Hubbard-Stratonovich transformation which allowed to solve the problem analytically for large sizes of matrices. We show that $\rho(E)$ depends crucially on the control parameter that drives the system through the transition between the ergodic and the fractal phases and it can be used as an order parameter.

# 1   Introduction

In recent decade the theory of Anderson localization discovered more than 60 years ago, experienced a certain revival. It is related first of all with the problem of Many-Body Localization (MBL) and absence of thermalization in quantum disordered systems [1, 2, 3]. The second motivation to revisit this problem was experimental realizations of Hamiltonians with long-range power-law hopping in dipolar cold-atom systems [4]. By placing such systems in optical cavity one may engineer the power-law hopping with the exponents that are variable in a broad range [5]. Both these problems have a common feature: in the problem of MBL the exponential smallness of the matrix elements connecting two basis states (e.g two bitstrings in a spin chain) at a given Hamming distance (the number of spin flips to get one bitstring from the other) can be compensated by the exponentially large number of such states; in the systems with power-law hopping on a lattice the polynomial smallness of the hopping matrix element between two sites at a large distance can be compensated by the polynomially large number of sites at this distance. This balance favors long-distance resonances even at small hopping which may result in the sparse, non-ergodic extended states. In contrast, in the short-range hopping systems on a finite-dimensional lattice, the number of sites grows polynomially but the effective hopping matrix element falls down exponentially at large distance. As the result, in a conventional single-body problem of localization on a lattice only few sites are typically in resonance (localization) at strong disorder or weak hopping; otherwise at large enough hopping all of them are in resonance to form an ergodic state. The intermediate, non-ergodic extended state may realize only at fine tuning of disorder strength, i.e. at the Anderson transition point. In contrast, in interacting systems and in system on a lattice with power-law hopping such states may form a *phase* that exists in some finite range of parameters.

The simple models for such systems are random matrix ensembles with independently fluctuating matrix elements with the distribution functions that depend on the distance from the main diagonal (see various examples in Ref. [6]). The simplest of such random matrix ensembles is the Rosenzweig-Porter (RP) ensemble [7], where *all* the off-diagonal matrix elements are independently and identically distributed (i.i.d.) according to a certain Gaussian distribution, while the diagonal ones are i.i.d. random Gaussian variables with a different variance. This ensemble is no longer invariant under the basis rotation which makes it a simple playground for studying the localization effects. In particular, one may make the variance of the diagonal matrix elements independent of the matrix size $N$, while the variance of the off-diagonal matrix elements being $N$-dependent $\propto N^{-\gamma}$ and small ($\gamma > 0$). A surprising property of this ensemble is that it exhibits, besides the localization transition at $\gamma = 2$, also a transition at $\gamma = 1$ from the ergodic phase identical to the one in the classic Wigner-Dyson random-matrix theory [8], to the non-ergodic extended phase with the fractal support of random eigenfunctions [9].

A further extension of the Rosenzweig-Porter ensemble is the non-Gaussian, strongly tailed distribution of *off-diagonal* matrix elements which higher moments are either fast growing with the matrix size (log-normal RP ensembles [10]) or they do not exist whatsoever (Lévy-RP [11]). Such models have a rich phase diagram in the space of two parameters, the *hopping parameter* $\gamma$ that determines the typical variance of the hopping matrix elements and *the tail parameter* ($p$ [10] or $\mu$ [11]) that controls the fat tails in the distribution of off-diagonal entries.

The local dynamics of these models is slow. In particular, the mean survival probability

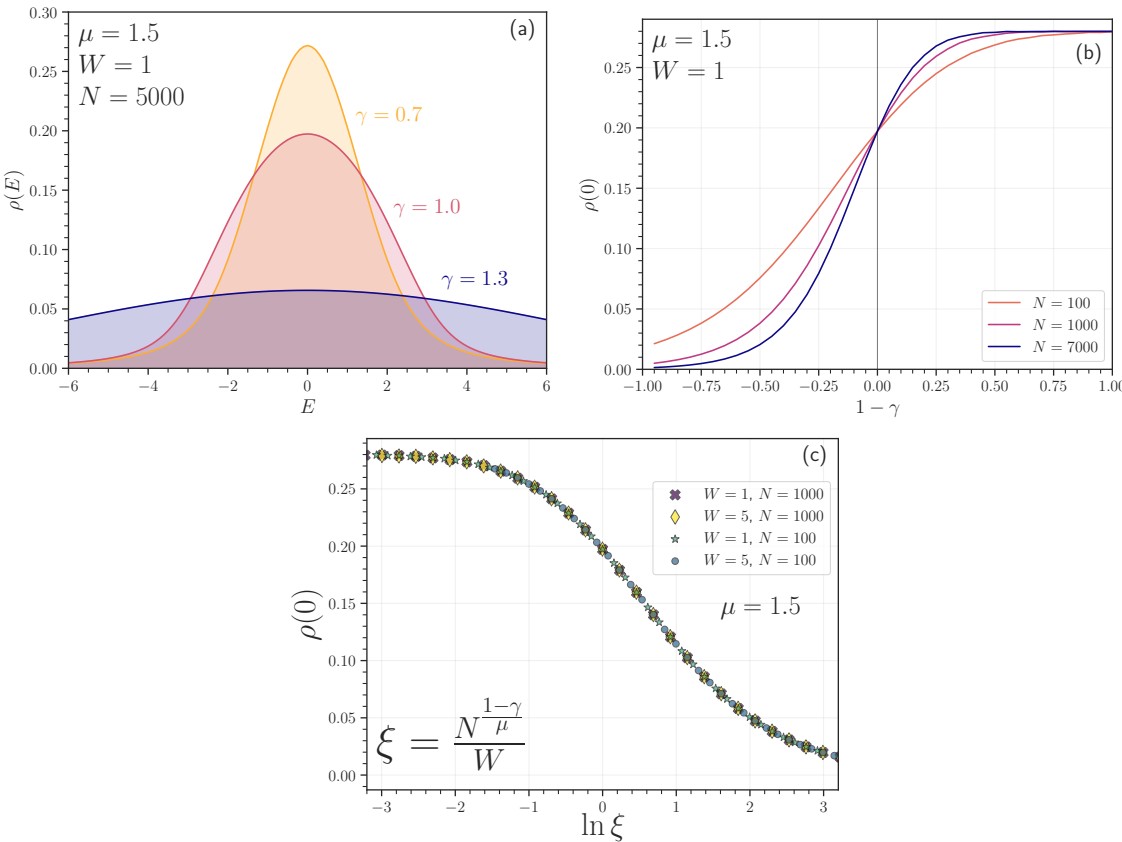

Figure 1: **DoS $\rho(E)$ at the ergodic transition for the Lévy-RP random matrices:**
(a) The diverging with $N$ band-width $B \propto N^{(1-\gamma)/\mu}$ in the ergodic phase, $\gamma = 0.7$, the $N$-independent DoS at the ergodic transition, $\gamma = 1$, and a convergent with increasing $N$ DoS in the non-ergodic extended phase $\gamma = 1.3$. (b) Dependence on $N$ of the "order parameter" $\rho(0)$ plotted as a function of $\tau = \gamma_c - \gamma$. (c) Single-parameter scaling: all curves for $\rho(0)$ as functions of $\ln \xi = \ln(W^{-1}N^{(1-\gamma)/\mu})$ for different disorder strength $W$ and different matrix sizes $N$ collapse to a single curve which depends only on $\mu$. All plots are obtained from our analytical results, Eqs.(31),(32).

demonstrates the *stretch-exponential* relaxation of the population of the single initially populated site [12]. This is reminiscent of the corresponding stretch-exponential relaxation in the Anderson model on Random Regular Graph [13, 10, 14] and in disordered spin chains [15, 16] which is associated with sub-diffusion.

It is important to emphasize that the family of Rosenzweig-Porter random matrix theories is principally different from the theories where *all* the entries are independently and identically distributed, for instance from the classic Wigner-Dyson RMT [8] and from the Lévy matrices [17, 18]. The presence of the *special diagonal* breaks the basis invariance and opens the way towards the localization. The difference in the distribution of diagonal and off-diagonal matrix elements makes possible existence of *non-ergodic extended* (fractal) states as a precursor of localization and of the Anderson localization itself.

The transition from the ergodic to the non-ergodic extended states (the *ergodic transition* at $\gamma = \gamma_{ET}$ [9, 6]) happens as a true phase transition in all the Rosenzweig-Porter models.

However, in contrast to the localization transition, it leads to a qualitative change of the *global* density of states (DoS) $\rho(E)$. Namely, in the non-ergodic phases the DoS in the thermodynamic limit tends to a distribution of diagonal entries which is $N$-independent, while in the ergodic phase $\rho(E)$ is strongly size-dependent.

We will show in this paper, using an analytic solution for the Lévy-RP model [11], that $\rho(0)$ may serve as an *order parameter* which tends to a non-zero value in the non-ergodic extended phase and tends to zero in the ergodic one. This evolution as a function of $\gamma - \gamma_{ET} = \gamma - 1$ depends on the disorder strength $W$ (proportional to the square-root of the variance of diagonal matrix elements) and on the matrix size $N$ combined in a *single parameter* $\xi = W^{-1} N^{(1-\gamma)/\mu}$ which corresponds to the finite-size scaling exponent $\nu = 1$ (see Fig.1).

To describe the finite-size scaling (FSS) at the transition from the ergodic to the extended non-ergodic phase one needs the solution for $\rho(E)$ at a finite system size $N$. This solution does not reduce to the solution [19] for the pure *Lévy ensemble* [17] in which all the matrix elements are identically distributed with the Lévy distribution. In this work we obtain an exact analytic solution for $\rho(E)$ at a finite (but large) matrix size $N$ using an extension of the Efetov supersymmetry (SUSY) approach [20].

One of the goals of the paper is to demonstrate how the SUSY formalism can be applied to the random matrices with the heavily tailed, strongly non-Gaussian distribution of off-diagonal entries combined with the different (Gaussian in our case) distribution of the diagonal ones. This method based on the *functional* Hubbard-Stratonovich transformation [21, 22] can be applied to compute other physical quantities for random Hamiltonians with heavily tailed distribution. The work on the correlation functions of global and local DoS and the mean survival probability is in progress and the corresponding results will be published elsewhere.

The rest of the paper is organized as follows. Section 2 provides a detailed description of the model studies and of the functional Hubbard-Stratanovich method which enables us to deal with heavy-tailed distributions using supersymmetric approach. It contains Eqs.(25),(27) as the main analytical result of the paper. Section 3 is devoted to scaling analysis of DoS behavior near ergodic transition at $\gamma = 1$. Section 4 demonstrates agreement of analytical expressions with results of direct diagonalization. Section 5 contains our conclusions.

# 2 Density of states for the Lévy and Lévy-RP matrices

## 2.1 Definitions

Our research object is $N \times N$ real symmetric matrix $\hat{H}$ which can be represented as the sum of two symmetric matrices:

$$\hat{H} = \hat{H}^{(1)} + \hat{H}^{(2)}, \tag{1}$$

where $\hat{H}^{(1)}$ is a diagonal random matrix with independent and identically distributed (i.i.d.) entries and $\hat{H}^{(2)}$ is a full matrix where *all* elements are i.i.d. The distribution of $\hat{H}^{(1)}$ and $\hat{H}^{(2)}$ are generally different. We consider two basic cases: (i) the case of the *Lévy matrices* [17, 18] where $\hat{H}^{(1)} = 0$ and $\hat{H}^{(2)}$ are independently and identically distributed according to the Lévy stable distribution [23, 24] and (ii) the case of the *Lévy-Rosenzweig-Porter (Lévy-RP) matrices* [11] where the entries of $\hat{H}^{(1)}$ are Gaussian distributed

$$P_1^{(W)} \left( H_{ii}^{(1)} \right) = \frac{1}{\sqrt{2\pi}W} e^{-\frac{H_{ii}^{(1)2}}{2W^2}} \tag{2}$$

and those of $H^{(2)}$ have a distribution:

$$P_2^{(\mu,\gamma)}\left(H^{(2)}{}_{ij}\right) = (N^\gamma)^{1/\mu}\, L_\mu\left((N^\gamma)^{1/\mu}\, H^{(2)}{}_{ij}\right), \tag{3}$$

where $L_\mu(x)$ is a symmetric Lévy stable distribution with the characteristic function:

$$\tilde{L}_\mu(k) \equiv \int_{-\infty}^{\infty} L_\mu(x)e^{-ikx}dx \equiv e^{-|k|^\mu}, \quad 0 < \mu \leq 2. \tag{4}$$

Notice that while the variance $W^2$ of $H_{ii}^{(1)}$ is independent of the matrix size $N$, the typical value of $H^{(2)}{}_{ij}$ scales with $N$ as $N^{-\gamma/\mu}$, and its variance diverges at $\mu < 2$ because of the tail in $L_\mu(x) \sim x^{-(1+\mu)}$. There are two special values of $\mu$: $\mu = 2$ where this tail disappears and the distribution $L_\mu(x)$ becomes Gaussian, and $\mu = 1$ when it coincides with the Cauchy distribution.

Using Eq.(4) we can find the characteristic function of rescaled $P_2^{(\mu,\gamma)}\left(H^{(2)}{}_{ij}\right)$ distribution:

$$\tilde{P}_2^{(\mu,\gamma)}(k,N) = \exp\left(-\frac{|k|^\mu}{N^\gamma}\right) \tag{5}$$

We would like to calculate mean density of state (DoS) $\rho(E)$ using Efetov's supersymmetric approach [20] further elaborated in Refs. [21, 22].

The partition function $Z(E,J)$ in terms of which the mean DoS $\rho(E)$ is found by differentiation over background field $J$, is given in this approach by the integral over the super-vectors $\phi_i$:

$$Z(E,J) = \int \prod_i [d\phi_i] \exp\left\{\frac{1}{2}\sum_{ij}\phi_i^\dagger\left[\left(E\hat{I} + J\hat{K}\right)\delta_{ij} - H_{ij}\right]\phi_j\right\} \tag{6}$$

$$\rho(E) = \frac{1}{2\pi N}\text{Im}\left.\frac{\partial\langle Z(E,J)\rangle_{1,2}}{\partial J}\right|_{J=0} \tag{7}$$

where

$$\phi_i = \begin{pmatrix} S_{i1} \\ S_{i2} \\ \chi_i \\ \chi_i^* \end{pmatrix}, \quad \phi_i^\dagger = \begin{pmatrix} S_{i1} & S_{i2} & \chi_i^* & -\chi_i \end{pmatrix} \tag{8}$$

is a super-vector with ordinary (commuting) $(S_{i1}, \quad S_{i2})$ and Grassmannian (anti-ommuting) $(\chi_i, \quad \chi_i^*)$ components, $\hat{K} = \text{diag}(1, 1, -1, -1)$ and $\hat{I}$ is identity matrix. We also will need a grassmannian integration rule:

$$\int \chi d\chi = \int \chi^* d\chi^* = \frac{i}{\sqrt{2\pi}} \tag{9}$$

## 2.2  Calculation process

We start by the averaging of the partition function (6) over the random entries of $\hat{H}$:

$$\langle Z(E,J)\rangle = \int \prod_i [\phi_i] \exp\left(\frac{i}{2}\sum_i \phi_i^\dagger\left(E + J\hat{K}\right)\phi_i + \sum_{i,j}\ln\left\langle\exp\left(-\frac{i}{2}H_{i,j}\phi_i^\dagger\phi_j\right)\right\rangle_{1,2}\right) \tag{10}$$

Notation $\langle ... \rangle_{1,2}$ means that averaging is done over both distributions (2) and (3). Since all $H_{ij}$ are not correlated one can split the sum into two terms (diagonal and off-diagonal). Also we use the fact that Lévy terms are typically $N^{\frac{\gamma}{2\mu}}$ times smaller than Gaussian ones and therefore can be neglected in the second term.

$$\sum_{i,j} \ln \left\langle \exp \left( -\frac{i}{2} H_{i,j} \phi_i^\dagger \phi_j \right) \right\rangle_{1,2} = \sum_{i \neq j} \ln \left\langle \exp \left( -\frac{i}{2} H_{i,j}^{(2)} \phi_i^\dagger \phi_j \right) \right\rangle_2 + \sum_i \ln \left\langle \exp \left( -\frac{i}{2} H_{i,i}^{(1)} \phi_i^\dagger \phi_i \right) \right\rangle_1$$

$$(11)$$

Let us first consider the averaging over the off-diagonal entries of $\hat{H}$. First of all independence of $H_{i,j}$ allows us to represent the average of products by the product of averages:

$$\sum_{i \neq j} \ln \left\langle \exp \left( -\frac{i}{2} H_{i,j}^{(2)} \phi_i^\dagger \phi_j \right) \right\rangle_2 = \ln \prod_{i < j} \left\langle \exp \left( -i H_{i,j}^{(2)} \phi_i^\dagger \phi_j \right) \right\rangle_2 = \frac{1}{2} \sum_{i \neq j} \ln \left\langle \exp \left( -i H_{i,j}^{(2)} \phi_i^\dagger \phi_j \right) \right\rangle_2$$

$$(12)$$

Using smallness $N^{-\gamma/\mu}$ of the typical off-diagonal matrix elements at large enough $N$, we represent:

$$\frac{1}{2} \sum_{i \neq j} \ln \left\langle \exp \left( -\frac{i}{2} H_{i,j}^{(2)} \phi_i^\dagger \phi_j \right) \right\rangle \approx \sum_{i \neq j} \frac{C \left( \phi_i^\dagger \phi_j \right)}{2N}$$

$$(13)$$

where we introduced the scalar function

$$C \left( \phi_i, \phi_j \right) \equiv C \left( \phi_i^\dagger \phi_j \right) = N \int dx P_2^{(\mu,\gamma)} (x, N) \left( e^{-i \phi_i^\dagger \phi_j x} - 1 \right).$$

$$(14)$$

Thus the averaging over the off-diagonal entries of $\hat{H}$ essentially reduces to the Fourier-transforming of the distribution function $P_2^{(\mu,\gamma)} (x, N)$.

It is important for further progress of calculations to have a simple enough characteristic function of $P_2^{(\mu,\gamma)}$. This was the reason to choose (among all distributions with the power-law tail cut at small values of $x$) the rescaled Lévy stable distribution for $P_2^{(\mu,\gamma)}(x)$. Using Eq.(5) for the corresponding characteristic function we obtain for $\gamma > 1$ and large enough $N$:

$$C(k) = -N^{1-\gamma} |k|^\mu$$

$$(15)$$

An obvious difficulty that still remains is the non-analytic power $\mu$ of $\phi_i^\dagger \phi_j$ in the functional after averaging over $\hat{H}_{ij}$ (instead of the quatric term arising for the Gaussian distribution). This non-analyticity encodes the fat tails in the distribution which, in their turn, determine the peculiar physical properties of the system. A related problem is that $C(\phi_i, \phi_j)$ couples the $\phi$- super-vectors at different sites $i$ and $j$.

In order to decouple the super-vectors we use *the functional Hubbard-Stratanovich(H-S) transformation* instead of the usual one (which converts the quatric term in the functional into a quadratic one that can be easily integrated). This non-trivial step was suggested (for different applications) in Refs.[21, 22] and rarely used since then. Since this mathematical trick is a common framework for treating all the random Hamiltonians with a fat tail in the distribution, we present it here in detail.

We start from the identity:

$$\exp\left(\frac{1}{2N}\int[d\psi]\,[d\psi']\,v\,(\psi)\,C\,(\psi,\psi')\,v\,(\psi')\right)=$$

$$\int Dg\exp\left(-\frac{N}{2}\int[d\psi]\,[d\psi']\,g\,(\psi)\,C^{-1}\,(\psi,\psi')\,g\,(\psi')+\int[d\psi]\,g\,(\psi)\,v\,(\psi)\right),\qquad(16)$$

where $C\,(\psi,\psi')$, $v(\psi)$ and $g(\psi)$ are some functions or fields.

We suppose that $C^{-1}$ operator exists and determined by the relation

$$\int[d\chi]\,C^{-1}\,(\phi,\chi)\,C\,(\chi,\psi)=\delta_{\phi,\psi}\qquad(17)$$

where $\delta_{\phi,\psi}$ is $\delta-$function in the space of supervectors. Choosing $v\,(\psi)=\sum\limits_{i=1}^{N}\delta\,(\psi-\phi_i)$ we will get an expression in which $\phi_i^\dagger$ and $\phi_j$ are decoupled for different $i,j$:

$$\exp\left(\sum_{i,j}\frac{C\left(\phi_i^\dagger\phi_j\right)}{2N}\right)=\int Dg\exp\left(-\frac{N}{2}\int[d\psi]\,[d\psi']\,g\,(\psi)\,C^{-1}\,(\psi,\psi')\,g\,(\psi')+\sum_i g\,(\phi_i)\right)$$
$$(18)$$

After substitution Eq.(18) into Eqs.(10)-(13) and averaging over $\hat{H}^{(1)}$, the partition function takes the form:

$$\langle Z\,(E,J)\rangle=\int Dg\exp\left(-\frac{N}{2}\int[d\psi]\,[d\psi']\,g\,(\psi)\,C^{-1}\,(\psi,\psi')\,g\,(\psi')\right.$$

$$\left.+N\ln\left\{\int[d\phi]\exp\left(\frac{i}{2}\phi^\dagger\left(E+J\hat{K}\right)\phi+g\,(\phi)+\ln\left(\int P_1^{(W)}\,(\epsilon)\,e^{-\frac{i}{2}\epsilon\phi^\dagger\phi}d\epsilon\right)\right)\right\}\right)\qquad(19)$$

Here we suppress the site indices in $\phi$ as after decoupling the integration over all $\phi_i$ are independent and identical, thus resulting in only the pre-factor $N$ in front of the result of integration over one of them denoted by $\phi$.

Performing the functional integration over $g$ by the steepest descent method (justified by a large pre-factor $N$ in the action), we get the following integral equation for $g$(for $J=0$):

$$g\,(\psi)=\frac{\int[d\phi]\,C\,(\psi,\phi)\exp\left(\frac{i}{2}\phi^\dagger E\phi+g\,(\phi)+\ln\left(\int P_1^{(W)}\,(\epsilon)\,e^{-\frac{i}{2}\epsilon\phi^\dagger\phi}d\epsilon\right)\right)}{\int[d\phi]\exp\left(\frac{i}{2}\phi^\dagger E\phi+g\,(\phi)+\ln\left(\int P_1^{(W)}\,(\epsilon)\,e^{-\frac{i}{2}\epsilon\phi^\dagger\phi}d\epsilon\right)\right)}\qquad(20)$$

This equation does not change under the unitary super-vector rotation $\phi\to\hat{T}\phi$, $\psi\to\hat{T}\psi$ since the r.h.s. of (20) contains only $\phi^\dagger\phi$ and $\psi^\dagger\phi$ combinations. For this reason the solution for $g(\phi)$ is invariant under this rotation. We therefore search for a solution $g\,(\phi)$ as a function of the invariant $\phi^\dagger\phi\equiv\mathbf{S}^2+2\chi^*\chi$.

The integral in the denominator of Eq.(20) has an integrand in which the super-symmetry is not violated. Thus due to the basic property of the super-symmetry method [20] it is equal to 1. Then introducing the components, Eq.(8), of a super-vector explicitly, we express $g\,(\phi)$ in a form

$$g\,(\phi)\equiv g_0\left(\phi^\dagger\phi\right)=g_0\left(\mathbf{S}^2\right)+2\chi^*\chi g_0'\left(\mathbf{S}^2\right)\qquad(21)$$

For the future calculation let us introduce a scalar function of commuting variables:

$$F\left(R^2\right) \equiv \frac{i}{2} E R^2 + g_0\left(R^2\right) + \ln\left(\int P_1^{(W)}\left(\epsilon\right) e^{-\frac{i}{2}\epsilon R^2} d\epsilon\right). \tag{22}$$

If we separate the commuting part from both sides of the integral equation Eq.(20) we obtain

$$g_0\left(\mathbf{S}^2\right) = -N^{1-\gamma} \int \frac{2dR_1 dR_2}{-2\pi} \left|\mathbf{SR}\right|^\mu e^{F(\mathbf{R}^2)} \frac{\partial F\left(R^2\right)}{\partial R^2} \tag{23}$$

Here we used the definition, Eq.(8), and the Grassman integration rule (9). After switching to the polar coordinates and integration by parts this expression takes the form:

$$g_0\left(x\right) = -x^{\mu/2} \frac{2N^{1-\gamma}}{\mu B\left(\frac{1}{2}, \frac{\mu}{2}\right)} \int_0^\infty dy \exp\left(F\left(y^{2/\mu}\right)\right), \tag{24}$$

where $B\left(\frac{1}{2}, \frac{\mu}{2}\right)$ is the $\beta$-function.

If we search the solution $g_0\left(x\right)$ in the form $g_0(x) \equiv -f_\mu(E) x^{\mu/2}$ we arrive at an integral equation for $f_\mu(E)$.

$$f_\mu(E) = \frac{N^{1-\gamma}}{B\left(\frac{1}{2}, \frac{\mu}{2}\right)} \int_0^\infty \frac{dy}{y^{1-\mu/2}} \exp\left(i\frac{E}{2}y - f_\mu(E)y^{\mu/2} + \ln\left(\int P_1^{(W)}\left(\epsilon\right) e^{-\frac{i}{2}y\epsilon} d\epsilon\right)\right) \tag{25}$$

Notice that this equation is not a true integral equation but rather an algebraic equation, as the function $f_\mu(E)$ is not integrated.

Now it is time to return to the DoS calculation. Since we know saddle-point solution of (20) we can write down the expression for $\rho(E) \propto \mathrm{Im} \frac{\partial Z}{\partial J}\Big|_{J=0}$ in the large-$N$ limit. Using (16) we obtain:

$$\frac{\partial \langle Z\left(E, J\right)\rangle}{\partial J}\Big|_{J=0} = \frac{iN}{2} \int [d\phi] \exp\left(\frac{i}{2} E \phi^\dagger \phi + g\left(\phi\right) + \ln\left(\int P_1^{(W)}\left(\epsilon\right) e^{-\frac{i}{2}\phi^\dagger \phi \epsilon} d\epsilon\right)\right) \phi^\dagger \hat{K} \phi \tag{26}$$

Performing grassmanian integration and using integration by parts we arrive at the final result

$$\rho\left(E\right) = \mathrm{Re}\left[\frac{1}{2\pi} \int_0^\infty dy e^{\frac{i}{2}Ey - f_\mu\left(\frac{E}{2}\right)y^{\mu/2} + \ln\left(\int P_1^{(W)}(\epsilon) e^{-\frac{i}{2}y\epsilon} d\epsilon\right)}\right] \tag{27}$$

where $f_\mu(E)$ should be extracted from Eq.(25).

Eqs.(25),(27) is the main result of this paper. It is valid in the limit of large (but finite) $N$ both for a pure Lévy matrices [17, 18] (corresponding to $W = 0, \gamma = 1$) and for the Lévy-RP matrices with the special diagonal. In particular, it works for the Lévy-RP matrices [11] with the Gaussian weight of diagonals, where

$$\ln\left(\int P_1^{(W)}\left(x\right) e^{-\frac{i}{2}tx} dx\right) = -\frac{W^2 t^2}{8}. \tag{28}$$

The result for the pure Lévy matrices is known in the mathematical literature [19]. In this case $\rho(E)$ is $N$-independent in the large-$N$ limit. It was obtained using the traditional mathematical tools that have nothing to do with the supersymmetric calculus used in the present paper.

The mean DoS for the more interesting case of Lévy-RP matrices [11] where there are both the localization and the ergodic transitions and a non-trivial fractal phase, is a totally new result. It allows to study a non-trivial and $N$-dependent variation of $\rho(E)$ as $\gamma$ crosses the ergodic transition at $\gamma = 1$.

Last but not least, the derivation in the framework of the Efetov's supersymmetric approach presented above contains elements common to all problems of random Hamiltonians with a heavy tailed distribution of parameters and thus it is quite general.

## 3 Single-parameter scaling for $\rho(E)$.

The DoS for the Lévy-RP matrices depends both on the strength $W$ of the diagonal disorder and on the typical value of the hopping (off-diagonal) matrix elements controlled by the parameter $\gamma$. However, at large enough matrix size $N$ this dependence is in fact a dependence of a single-parameter:

$$\xi = W^{-1} N^{\frac{1-\gamma}{\mu}}. \tag{29}$$

To show that one can make a rescaling

$$\begin{aligned}
E &\rightarrow \epsilon N^{\frac{1-\gamma}{\mu}}, \\
f_\mu(E) &\rightarrow Y_\mu(\epsilon) N^{\frac{1-\gamma}{2}}, \\
y &\rightarrow t N^{-\frac{1-\gamma}{\mu}}.
\end{aligned} \tag{30}$$

Then Eqs.(25),(27) take the form:

$$Y_\mu(\epsilon) = \frac{1}{B\left(\frac{1}{2}, \frac{\mu}{2}\right)} \int_0^\infty \frac{dt}{t^{1-\mu/2}} \exp\left(\frac{i}{2}\epsilon t - Y_\mu(\epsilon) t^{\mu/2} - \xi^{-2} \frac{t^2}{8}\right) \tag{31}$$

$$\rho(\epsilon) = \text{Re}\left[\frac{1}{2\pi} \int_0^\infty dt \exp\left(\frac{i}{2}\epsilon t - Y_\mu(\epsilon) t^{\mu/2} - \xi^{-2} \frac{t^2}{8}\right)\right]. \tag{32}$$

Now it is clear that:

- in all the cases $\rho(E)$ depends on a single parameter $\xi$, Eq.(29), but not on $N$ and $\gamma$ separately [1]

- For Lévy matrices (corresponding to $W = 0, \gamma = 1$) $\rho(E)$ is an $N$-independent function of the energy $E$.

- For Lévy- RP matrices ($W \neq 0$) and $\gamma < 1$ (the ergodic phase) the DoS $\rho(E) = N^{-\frac{(1-\gamma)}{\mu}} \rho_{\text{Lévy}}\left(E N^{-\frac{(1-\gamma)}{\mu}}\right)$

  converges in $N \rightarrow \infty$ limit to that for the Lévy matrices with the rescaled energy, while for $\gamma > 1$ (fractal or localized phase) it converges to the Gaussian distribution of the diagonal matrix elements. At the ergodic transition $\gamma = 1$ $\rho(E)$ is $N$-independent but depends on the diagonal disorder $W$ and $\mu$.

---

[1] For the Gaussian Rosenzweig-Porter ensemble this result was obtained recently by the replica method [25].

The inflation of the body of the distribution (and the corresponding decrease of $\rho(0)$ by normalization) in the ergodic phase of Lévy-RP model is illustrated in Fig.1(a,b). The single-parameter scaling is reflected in the perfect collapse of data for different $N$ and $\gamma$ in Fig.1(c).

The single-parameter scaling allows to find the critical exponent $\nu$ of finite-size scaling (FSS) at the ergodic transition $\gamma = 1$. By definition of FSS any quantity, e.g. $\rho(0)$, near the $\gamma$-driven transition must obey at $|\gamma - 1| \ll 1$ the scaling relation:

$$\rho(0) = R_\mu \left( L^{\frac{1}{\nu}} (\gamma - 1) \right), \tag{33}$$

where $R_\mu(x)$ is the scaling function that depends only on $\mu$, and $L$ is a properly defined length scale. For all Rosenzweig-Porter matrices $L = \ln N$. In a particular case of Lévy-RP matrices the single parameter scaling suggests that the dependence is only on $\xi = \exp[\ln N (1 - \gamma)/\mu]$, which implies that the function $R_\mu$ depends on the combination $L(1 - \gamma)$, where $L = \ln N$, and on $\mu$. From that it immediately follows that the exponent $\nu$ at the ergodic transition is equal to:

$$\nu = 1. \tag{34}$$

# 4 Limiting cases and numerical verification

## 4.1 Pure Lévy ensemble

In this section we verify our analytical result, Eqs.(25),(25), by exact numerical diagonalization and averaging over the ensemble of corresponding random matrices. We start by the case of Lévy matrices that corresponds to $\gamma = 1, W = 0$.

A general receipt is to solve Eq.(25) numerically, then substitute it in Eq.(27) and compare the result with the result of numerical diagonalization of random matrices. However in some trivial cases Eqs.(25), (27) allow for an analytical solution. In particular, the case $\mu = 2$ represents the textbook example of the Gaussian Orthogonal Ensemble (GOE) [8]. In this case Eq.(25) reduces to the quadratic equation with the solution for $\rho(E)$ in a form of celebrated semi-circle [2]

$$f_2(E) = \frac{1}{2} \left( i\frac{E}{2} \pm \sqrt{2 - \frac{E^2}{4}} \right) \Rightarrow \rho_{\text{GOE}}(E)\bigg|_{\mu=2} = \frac{1}{2\pi} \sqrt{2 - \frac{E^2}{4}} \tag{35}$$

For an arbitrary $0 < \mu < 2$ some analytical results are also possible. In particular, it could be useful to calculate $\rho_{\text{Lévy}}(E = 0)$:

$$f_{\text{Levy}}(\mu, 0) = \sqrt{\frac{2}{\mu B \left( \frac{1}{2}, \frac{\mu}{2} \right)}} \Rightarrow \rho_{\text{Levy}}(0) = \frac{\Gamma \left( \frac{2}{\mu} + 1 \right)}{2\pi} \left[ \frac{\mu B \left( \frac{1}{2}, \frac{\mu}{2} \right)}{2} \right]^{1/\mu} \tag{36}$$

However, the goal of this section is to compare the analytical results with the results of numerical diagonalization, in order to establish how well the saddle-point approximation, Eq.(20), works at a reasonably large matrix sizes $N \sim 5000$. One can see the accuracy of this approximation in Fig.2.

---

[2]Lévy stable distribution at $\mu = 2$ reproduces the normal distribution with variance $\sigma = \sqrt{\frac{2}{N}}$ This is why the eigenvalues are normalized differently and one can expect the spectra from $-2\sqrt{2}$ to $2\sqrt{2}$ instead of $(-2, 2)$ in case of normal distribution with $\sigma = \sqrt{\frac{1}{N}}$.

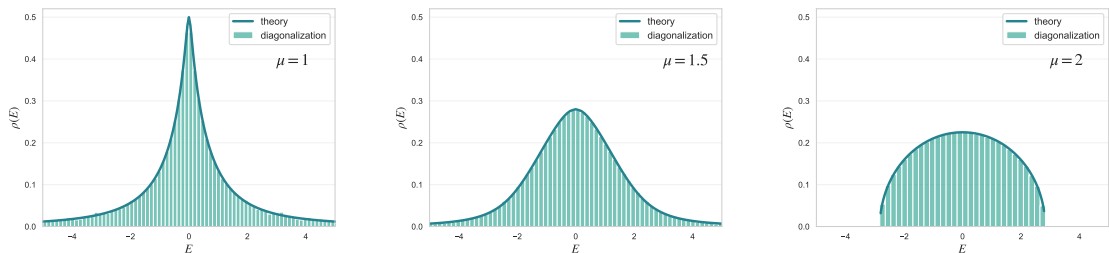

Figure 2: Comparison of the analytical results of Eqs.(25),(27) (shown by green lines) with the numerical diagonalization of the Lévy matrices for $N = 5000$(10 different samples) and $\mu = 1$(Cauchy), $\mu = 3/2$, $\mu = 2$(Gauss) (shown by histograms).

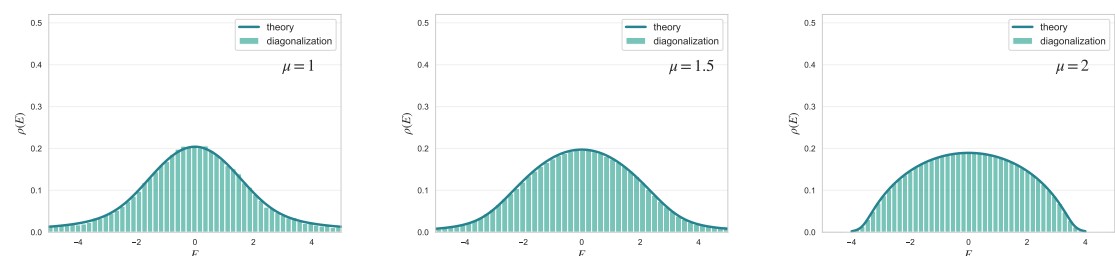

Figure 3: Comparison of the analytical results of Eqs.(25),(27) (shown by green lines) with the numerical diagonalization of the Lévy-RP matrices for $\gamma = 1$, $W = 1$, $N = 5000$(10 samples) and $\mu = 1$, $\mu = 1.5$, $\mu = 2$ (shown by histograms).

## 4.2   Lévy-RP ensemble

Now we perform a similar comparison for the Lévy-RP matrices at $\gamma = 1$ and $W = 1$. The choice of $\gamma = 1$ is the most non-trivial case, as for $\gamma > 1$ the $N \to \infty$ limit of $\rho(E)$ coincides with the Gaussian distribution of the diagonal elements, while for $\gamma < 1$ it coincides with the rescaled $\rho(E)$ for pure Lévy matrices. The results for the three values of $\mu$ ($\mu = 2$, $\mu = 1$ and $\mu = 3/2$) are shown in Fig.3. As for pure Lévy matrices, the comparison demonstrates an excellent coincidence of the analytical results and those of numerical diagonalization. Also clear is the hybrid character of the distribution: for $\mu = 2$ the band edge is no longer sharp as for a semi-circle, with appearance of the Gaussian tails; for $\mu = 1$ the cusp at $E = 0$ is rounded as for the Gaussian distribution.

## 5   Conclusion

In this work we show how to obtain spectral statistics of random matrices with heavily tailed distribution of elements within the Evetov's super-symmetry formalism. We consider two important examples: the pure Lévy symmetric matrices where all the elements are i.i.d. with the Lévy $\mu$-stable distribution and the Lévy-Rosenzweig-Porter matrices, where the diagonal elements are i.i.d. with the Gaussian distribution and the off-diagonal elements are i.i.d. with the Lévy $\mu$-stable distribution and a small typical value that scales with the matrix size $N$ as

$\sim N^{-\gamma/\mu}$. The fact that the diagonal matrix elements are $\sim N^0$ and the off-diagonal elements are typically small for large matrices results in a rich phase diagram with the ergodic, fractal and the localized phases and transitions between them. By computing the mean spectral density we show that it is sensitive to the transition between the ergodic and the fractal non-ergodic states (the ergodic transition), with the maximal spectral density $\rho(0)$ behaving like an order parameter for such a transition. Furthermore, we have shown that the dependence of $\rho(0)$ on the matrix size and the strength of disorder reduces to the dependence on a single parameter $\xi$. From the dependence of this parameter on the matrix size $N$ and the control parameter $\gamma$ that drives through the ergodic transition we found that the critical exponent of the finite-size scaling for this transition is $\nu = 1$.

All the analytically obtained results are verified by exact numerical diagonalization of matrices of large sizes.

The mathematical formalism employed in computing analytically the mean spectral denisty within the Evetov's super-symmetry approach is quite general and can be applied to different problems of random Hamiltonians which parameters have broad distributions with heavy tails.

## Acknowledgments

We are grateful to Yan Fyodorov and Alexei Lunkin for numerous useful discussions. Author also thanks Andrei Safonov for help with numerical simulations.

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
