# Peer review of "Spectral properties of L\'evy Rosenzweig-Porter model via supersymmetric approach"

_SciPost Physics_

## Round 1 · Referee Report · Anonymous (Referee 1) · 2024-9-27

Strengths

New and interesting results, obtained through a compact and powerful calculation (the method however is expressly not new).

Weaknesses

More attention should be paid to the overall presentation.

Report

In this work, the authors address the spectral density of the Lévy and Lévy-Rosenzweig-Porter model. This and several other variations of the RP random matrix ensemble have lately received a lot of attention, in a renewed effort to model key aspects of many body localization (MBL).
The calculation is performed within the supersymmetric formalism, and the result is shown to depend on a single scaling parameter, thus making the spectral density itself sensitive to the transition between the ergodic and the fractal phase of the model (first studied in Ref. [11]). 
All the result are tested against direct numerical diagonalizations.

I believe that this manuscript contains new and interesting results, and also that the functional Hubbard-Stratonovich transformation employed in the calculation has indeed not received all the attention it would deserve (even though examples of its use can in fact be found in the recent literature, see below).
In my view, this work may be suitable for publication in SciPost Physics — however, I have first a few remarks that I would like the authors to address, as I detail below. Most of them concern the overall presentation of the results, given also the generic nature of the audience of this journal.

Requested changes

  1. I feel that the presentation could be greatly improved with a relatively small effort, so as to make the paper more accessible to non-specialists. I am referring in particular to the introduction: for instance, the sentence “This balance favors long-distance resonances even at small hopping which may result in the sparse, non-ergodic extended states” is virtually incomprehensible for a non-specialist, since “resonance” and “non-ergodic state” have not yet been defined, even vaguely. 
 I am fully aware that it is standard, in the field in MBL, to adopt such terms without further ado. However, I think that an eccessive use of jargon unnecessarily makes the whole field harder to access for new scientists, and often produces ambiguities that could be very easily avoided.
 Besides my personal opinion, this is in fact the first general acceptance criterium of SciPost Physics.

  2. I found several misprints along the manuscript. For example,

  3. Page 2, “matrix elements which higher moments” -> “matrix elements whose higher moments”
  4. Page 5, under Eq. (8), “anti-ommuting”

  5. Page 6, under Eq. (15) “quatric” -> “quartic” (2 occurrences)

  6. Page 10, under Eq. (33), “is the scaling function” -> “is a scaling function”.

  7. Punctuation is missing at the end of most displayed equations, which makes the reasoning harder to follow.

  8. In Eq. (11), the diagonal Lévy terms are discarded on the basis of their being subleading (for large $N$) with respect to the Gaussian terms. While this may be legit, I doubt whether it is necessary — they might as well be taken into account with limited modifications, e.g. possibly an extra factor $\sqrt{2}$ with respect to the off-diagonal terms. 
This is not entirely pedantic, since later in the paper the Lévy limit is explicitly considered, and in that case the Gaussian diagonal terms are not there at all.

  9. At the end of page 6, “This non-trivial step was suggested (for different applications) in Refs.[21, 22] and rarely used since then.” While I agree that the functional HS transformation is an underestimated computational tool, I also think that it would be relevant to cite here some recent examples of its use for closely related problems: see for instance [10.1088/1751-8121/acdcd3] and [arXiv:2408.10530v1]. The first of these two papers addresses the spectrum of sparse random matrices with row constraints, using the very same supersymmetric technique employed here.

  10. On page 7, under Eq. (20), I noted that the super-vector rotation $\hat T$ was not defined anywhere in the manuscript. Although it can be found e.g. in Mehta’s book, I think it’s important to include it here explicitly, to ensure reproducibility.

  11. On page 8, I appreciate that Eq. (25) is not an integral equation, but it’s excessive to state that it is algebraic. Even a simple trascendental equation such as $x=\sin(x)$ cannot be called algebraic.

  12. I’ve been initially puzzled by the fact that the “spectrum of the sum of two random matrices” is a well-studied problem that can be solved within free probability (aka the “Zee formula”), once the resolvent (or Cauchy-Stieltjes transform) of the two individual matrices is known. Now, the resolvent of a diagonal matrix with i.i.d. entries is easily found, while the resolvent associated to a Lévy matrix was in principle studied already in Ref. [17] of the present manuscript. 
 However, to the best of my understanding (this is after all a mathematical paper), Ref. [17] does not contain the resolvent of a Lévy matrix in closed form. I’ve searched through the recent literature, and concluded that an explicit expression of such resolvent indeed is yet to be found — which makes the free probability prescription inapplicable to the Lévy-RP case. 
It may be worth highlighting this, since it makes the results of this paper even more remarkable (but of course it’s up to the authors to decide).

  13. In general, the mere knowledge of the spectrum of a random matrix is not a sufficient proxy of the nature of its eigenstates — one rather needs additional knowledge of two-point functions and/or inverse participation ratios. 
In this manuscript, the newly found spectral density of the Lévy-RP ensemble exhibits a qualitative change in correspondence of $\gamma=1$, from which the authors deduce that ”the mean spectral density […] is sensitive to the transition between the ergodic and the fractal non-ergodic states”. However, it seems to me that the whole claim is convincing only in view of previous works on the phase diagram of the model, notably Ref. [11], where $\gamma=1$ was shown to mark the onset of non-ergodic states. 
I simply suggest to the authors to recall e.g. in Section 2 these basic facts about the phase diagram of the Lévy-RP ensemble, rather than simply assuming that the reader is already familiar with them.

Recommendation

Ask for major revision

  • validity: high
  • significance: good
  • originality: ok
  • clarity: ok
  • formatting: below threshold
  • grammar: below threshold

Author:  Mikhail Feigel'man  on 2024-10-04  [id 4834]

(in reply to Report 1 on 2024-09-27)

Please see detailed reply in the attached file

Attachment:

Answer_to_Referee_I_TU1Uqvu.pdf

Anonymous on 2024-10-21  [id 4880]

(in reply to Mikhail Feigel'man on 2024-10-04 [id 4834])

I thank the authors for carefully addressing all the issues that I had raised in my report, and I am happy to recommend publication of the revised manuscript in SciPost Physics.

---

## Round 1 · Referee Report · Anonymous (Referee 2) · 2024-10-28

Strengths

see report

Weaknesses

see report

Report

The paper describes the application of the supersymmetric approach to derive the density of states for random matrix ensembles of the Rosenzweig-Porter type, with off-diagonal elements distributed according to large Lévy-distributions with fat tails. These ensembles were recently introduced to explore the nature of the non-ergodic delocalized phase—characterized by fractal fluctuations and distinct from both the ergodic delocalized and Anderson localized phases—in cases where off-diagonal elements follow fat-tailed distributions. Notably, for log-normal distributions, which closely resemble random regular graphs in effective infinite dimensions, it was found that the non-ergodic delocalized phase is absent, resulting instead in a tri-critical point between the ergodic and localized phases.
In this article, the authors derive the average density of states for such matrix ensembles analytically and find that it crucially depends on system size, exhibiting distinct behavior across the ergodic to non-ergodic transition. The main contributions of the paper are both methodological and qualitative. The methodological advance is the implementation of a functional Hubbard-Stratonovich transformation, necessary to account for the large fluctuations of matrix elements. The key qualitative finding is that, in this type of random matrix, the average density of states acts as an order parameter for the ergodic to non-ergodic transition. This result is somewhat surprising, as in Anderson transitions, the average density of states is typically insensitive to the transition.
The paper could benefit from some improvements, specifically by adding discussions and comparisons to other systems, which would clarify the generality or specificity of the authors’ findings and methods. Below, I list some suggestions, not in order of importance:
• An important methodology in this context is the cavity approach, which is exact for tree graphs. A relevant paper on this approach, Bogomolny and Giraud, PRE 88, 062811, provides analytical calculations of the average density of states, citing additional references. The authors of this paper explore approximations and compare them to exact numerical results. Since the cavity approach may be applicable to the random matrix ensembles under consideration and the log-normal Rosenzweig-Porter model relates to random regular graphs, it would be interesting if the authors could compare their approach and results with those of the cavity approach. Specifically, could the authors describe in some detail the case corresponding to random regular graphs (RRG)?
• In describing the method, it would be helpful if the authors clearly explained why the standard Hubbard-Stratonovich (HS) transformation is not suitable in this context and specified the cases in which the functional HS approach is applicable.
• This class of random matrix ensembles is unusual in that it depends explicitly on matrix size, suggesting that some size-dependent properties might be unique to this model. Given that one of the authors' main motivations relates to many-body localization and its multifractal properties in Hilbert space, how do the findings compare with known results for models displaying signs of a many-body localization transition?
• Similarly, can the authors discuss finite-size effects on the density of states in comparison to those seen in other random matrix models?
• In Anderson transitions, the density of states plays a crucial role when it vanishes, typically at the band edges. Is this the case here, and is there a mobility edge?

Requested changes

see report

Recommendation

Ask for minor revision

  • validity: top
  • significance: high
  • originality: high
  • clarity: good
  • formatting: good
  • grammar: good

Author:  Mikhail Feigel'man  on 2024-10-30  [id 4917]

(in reply to Report 2 on 2024-10-28)

Please see detailed reply from the authors in the attached file

Attachment:

Answer_to_Referee_II.pdf

Anonymous on 2024-11-08  [id 4948]

(in reply to Mikhail Feigel'man on 2024-10-30 [id 4917])

The authors have clarified the points I raised and I think the new version of the manuscript should be published in SciPost.

---

## Editorial Decision

resubmitted